# Identifying Gaps in the Investigation of the Vredefort Granophyre Dikes: A Systematic Literature Review

**Matthew S. Huber** \*[ID] **and Elizaveta Kovaleva**[ID]

Department of Geology, University of the Free State, Bloemfontein 9300, South Africa; kovalevae@ufs.ac.za
\* Correspondence: huberms@ufs.ac.za

**Abstract:** The Vredefort impact structure is among the oldest and largest impact structures preserved on Earth. An understanding of its key features can serve as a guide for learning about the development of basin-sized impact structures on Earth and other planetary bodies. One of these features is the so-called Vredefort granophyre dikes, which formed when molten material from the impact melt sheet was emplaced below the crater floor. The importance of these dikes has been recognized since the earliest studies of the Vredefort structure, nearly 100 years ago. The present study is a systematic literature review to determine the extent to which peer-reviewed scientific publications have generated unique data regarding the granophyre dikes and to investigate how scientific methods used to investigate the granophyre have changed over time. In total, 33 unique studies have been identified. Of those, more studies have been performed into the core-collar dikes than the core dikes. The majority of the studies have focused on field analyses, bulk geochemistry, and the studies of mineral components. The granophyre has long been recognized as a product of post-deformational processes and thus has been a target of age dating to constrain the minimum age of the impact event. In the last 25 years, studies of stable isotopes and shock deformation of minerals in lithic clasts within the dikes have taken place. A small number of geophysical studies relevant to the granophyre dikes have also been undertaken. Overall, there has been a relatively small number of studies on this important rock type, and the studies that have taken place tend to focus on two particular dikes. Several of the dikes have only been investigated by regional studies and have not been specifically targeted. The use of modern techniques has been lacking. More fieldwork, as well as geophysical, isotopic, microstructural studies, and application of novel techniques, are necessary for the granophyre dikes to be truly understood.

**Keywords:** impact crater; melt rock; planetary process

## 1. Introduction

Impact cratering is a fundamental process in the Solar System, with impact craters being abundant on the surfaces of the vast majority of terrestrial bodies that have been observed (e.g., [1]). The process of impact cratering results in such effects as structural deformation, shock metamorphism, melting, and vaporization of the target materials [2]. The impact process can be better understood by studying these effects.

Basin-forming impact events cause sufficient melting to form voluminous melt sheets that can be up to several kilometers thick [2]. On terrestrial bodies, such as the Moon or Mars, large multi-ring basins with melt sheets are selected for scientific exploration missions (e.g., [3]). On Earth, the three largest preserved impact structures are (from the youngest to the oldest): the Chicxulub, Sudbury, and Vredefort structures; all three formed significant volumes of impact melt [4]. The three structures have a different level of exposure and preservation state, which makes them complementary to each other in terms of developing our understanding of impact events, and allows us to study different aspects of

multi-ring basins. The upper portions of the Chixculub impact crater are not exposed on the surface, but the nature of the impact melt is being explored through scientific drilling [5]. The Sudbury melt sheet has been the target of significant economic exploration (e.g., [6]). In contrast, the melt sheet of the multi-ring Vredefort impact structure has been erosionally removed, with only a few impact melt dikes preserved today within the exposed basement of the target (e.g., [7]).

The Vredefort impact structure (Figure 1) is centered ~120 km south-southwest of Johannesburg, South Africa (e.g., [8]), in the middle of the Witwatersrand goldfields. The exposed diameter of the "Vredefort Dome" is approximately 60 km along its north-south axis. However, the "Dome" represents only the central uplift of the Vredefort structure. The latter also includes the Potchefstroom Synclinorium and has at least partial structural control on the Witwatersrand goldfields (e.g., [9–11]). Based on the geophysical modeling [12], the diameter of the original impact structure was estimated to be ca. 300 km, whereas numerical modeling has suggested an initial diameter of ca. 170–180 km [13,14]. In either case, the Vredefort structure is among the three largest impact structures on Earth, along with the Chicxulub and Sudbury impact structures [4].

The core of the central uplift of the Vredefort structure is composed of Archean granitoids that were uplifted from the middle crust by the impact [10,15]. Surrounding the core is a collar of the central uplift that includes strata of the Dominion Group, the Witwatersrand, Ventersdorp, and Transvaal Supergroups [16]. These collar rocks form a series of semicircular ridges in the west and northwest parts of the Dome and are overturned by the impact from their initial horizontal position by dipping angles of 90–130°, being rotated outwards [17]. The southeast part of the structure is covered by Karoo Supergroup sedimentary rocks and is not well exposed, except for an inlier known as the Greenlands Formation [18]. The age of the Vredefort structure is 2020 ± 2 Ma (e.g., [19,20]; see below), making it the second oldest confirmed impact structure on Earth. The Vredefort structure is a unique basin-sized impact structure due to its deep erosional level, with an estimated ca. 10 km of erosion that had taken place since its formation [21,22].

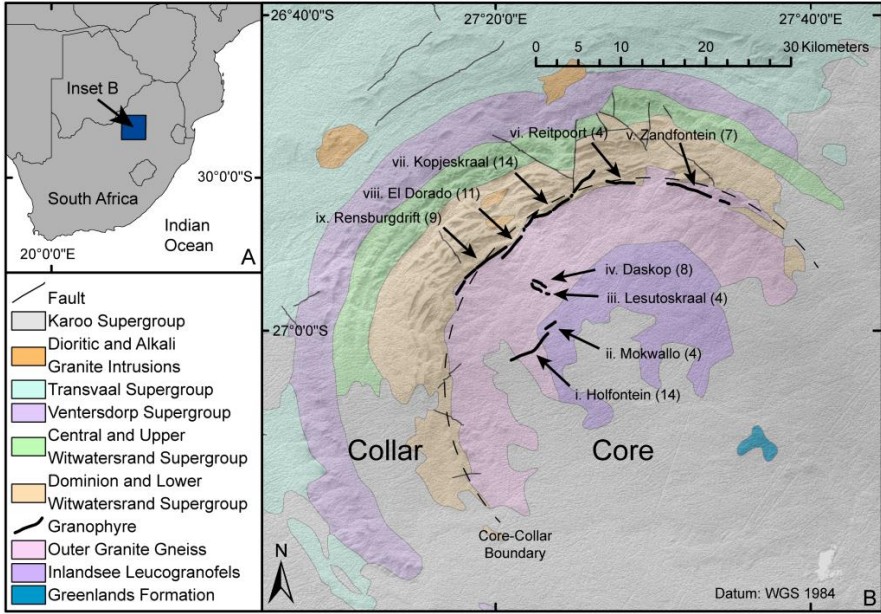

**Figure 1.** Geological map of the Vredefort Dome. The granophyre dikes are shown in bold lines, with the roman numeral indicating the designation of the dikes as given by [23], the name of the dikes as used in this study, and previous literature, and the number of studies on the respective dike in brackets. The dashed line shows the approximate boundary between the core and the collar of the central uplift. Topography generated from ASTER GDEM, which is a product of the Ministry of Economy, Trade, and Industry (METI) and the National Aeronautics and Space Administration (NASA).

Many lines of evidence have shown the impact genesis of the Vredefort structure. The unusual origin of the structure from a singular point was recognized by early workers [24], although this interpretation was not matched with the concept of an extraterrestrial impact event; rather, most of the early workers referred to either the "enigmatic origin" or pointed to a "cryptoexplosion" as a formation mechanism. The convincing case of an impact origin was first made by Dietz [25], who argued based on the presence of shocked minerals, shatter cones, and unusual melt rocks (although discussions of a cryptoexplosion continued up until the early 2000s). Two major types of impact-generated melt rocks have been recognized: pseudotachylites, an in situ melt rock (melt breccia) named for its similarity to the volcanic glass—tachylyte, and the Vredefort granophyre, an ex situ melt rock named for the granophyric texture of its groundmass [16]. Pseudotachylites (in some studies indicated as "pseudotachylitic breccia" [26]) have been observed in all lithologies of the Vredefort structure, including numerous locations within the Witwatersrand goldfields (e.g., [17,27]). In contrast, the granophyre is observed as nine dikes in the core and at the core-collar boundary of the Vredefort Dome (Figure 1). Various studies have investigated the properties of these two melt rock types, with the conclusion that they both originated during an impact event but through distinct mechanisms (e.g., [28,29]).

The Vredefort granophyre is generally understood to be derived from the impact melt sheet. The melt sheet was formed as a result of the bulk melting of the upper crust during the impact process. After the melt sheet formation, the impact melt was emplaced below the crater floor via deep crustal fractures and formed granophyre dikes (e.g., [30,31]). The granophyre was likely emplaced at the latest stages of impact crater development and thus is undeformed [25,32]. In contrast, there are multiple proposed mechanisms to explain the formation of pseudotachylite. Pseudotachylites form as a result of either shock-related ultracataclasis and melting and/or frictional melting coincident with the passage of the shock wave during the formation of the impact structure [33], from shock release [34], or in the postimpact modification stage by an equivalent mechanism to tectonic pseudotachylites [35]. The understanding of pseudotachylites is complicated because they can also form in tectonic settings [36], and tectonic pseudotachylites are also present in the Vredefort basement rocks [37]. The equivalents of granophyre and pseudotachylite have been observed in other impact structures (e.g., [38–40]).

Due to the large size and complex nature of the Vredefort impact structure, it is best understood in terms of its component features. The Vredefort granophyre is a critical lithology for study due to its suggested connection to the eroded impact melt sheet of the Vredefort structure. Many key aspects of the Vredefort impact event can only be determined through the study of the granophyre, such as identification of the impactor composition, evolution of the impact structure in time (e.g., the fracturing of the crater floor), the evolution of the impact melt, properties of the upper-level impactites, whose fragments are preserved within the dikes, or any other property related to the formation of Vredefort impact melt sheet.

Previous review papers have discussed the overall geology of the Vredefort structure or the pseudotachylites in particular (e.g., [8,34,41]), but these have not been systematic reviews. There has been one review paper concerning pseudotachylites and granophyre [8], but this review was written prior to the publication of ca. 40% of the literature concerning the granophyre. In this study, we present a systematic review of all of the geological literature concerning the Vredefort granophyre to identify what data have been generated and communicated through the scientific literature, determine what analytical trends are present, and highlight gaps in the knowledge to guide future studies.

## 2. Materials and Methods

A systematic literature review was carried out to identify what data have been generated regarding the Vredefort granophyre dikes. A systematic literature review is an effort to identify, appraise, and synthesize all empirical evidence that meets particular eligibility criteria to answer a specific research question [42]. The systematic search and categorization of studies, with defined, repeatable criteria for determining which studies should be included, has been shown to be unbiased and reproducible [43–45].

The major use of unbiased search methods is to determine areas of uncertainty and identify gaps in research [46]. The review criteria are based upon three steps: (1) a literature search; (2) selection of relevant studies; (3) categorization, and synthesis of the findings. Although systematic literature reviews were originally developed for medical science, they have since been utilized in a broad range of scientific fields, including geoscience (e.g., [47]), to determine the scope of work that has been done in an area of research and to identify gaps in research. A systematic review differs from traditional reviews that typically provide an overview of a topic without explicit methodology to determine which studies are included [42]. As a result, such reviews might be—even unintentionally—biased towards representing a particular school of thought or group of researchers. A systematic literature review does not favor any particular group, and the results can be repeated independently.

To guarantee the robustness of the present review, only peer-reviewed publications were taken into consideration, so that comments and replies, editorials, and postgraduate dissertations and theses were excluded. Review papers were not included, as they do not involve the generation of new data concerning the dikes. Only papers published in English were selected. The search was carried out in the online databases Scopus, Web of Science, and ScienceDirect. The controlled vocabulary keyword used for the search was "Vredefort Granophyre," to exclude studies on granophyre unrelated to Vredefort. Papers that include these terms in the title, abstract, topic heading, or keywords were considered in the review process. The search was carried out in May–June 2020. The literature search produced a total of 119 results: 28 in Scopus, 51 in Web of Science, and 40 in Science Direct. Duplicated results were filtered, so that only 53 unique results remained, whose abstracts were individually screened. In this process, five comments and four review papers were removed. The full texts of the remaining 44 papers were then reviewed in-depth. Of these, studies that did not include a new analytical report concerning the granophyre were removed. In this way, 21 studies were found to be unrelated to granophyre (primarily studies that investigated other aspects of Vredefort), and two studies that investigated granophyre using literature data without producing a new analysis were removed. By examining the bibliographies of the remaining papers, an additional 12 related studies were identified and scrutinized. A total of 33 research papers were, therefore, selected for the present review. (Notably, although we did not include postgraduate theses or non-English publications within our criteria, we observed less than five such documents while performing our review. It is possible that other such documents exist, but have not been cited by later workers.)

Our selection thus represents the total number of unique peer-reviewed scientific works in English that have published the results of fieldwork and/or laboratory analysis of the Vredefort granophyre rocks. To further analyze papers, Table 1 was produced, which included information concerning the authors, year of publication, journal title and discipline, and the geographic location of each study (i.e., specific granophyre dike). The papers were also categorized based on their aims and topics, the research methods applied, and their main results.

**Table 1.** Results of the literature search for "Vredefort Granophyre." Papers are listed chronologically. Numbers in brackets after the dike names indicate the numbering used by Therriault et al. [23].

| | Researchers | Year of Study | Publication | Field Studies | Bulk Geochemistry | Mineral Components | Geochronology | PGE and Isotope Geochemistry | Geophysics | Core Dikes | Core-Collar Dikes | Specific Dike(s) |
|---|---|---|---|---|---|---|---|---|---|---|---|---|
| 1 | Hall and Mollengraff [24] | 1925 | Koninklijke Akademie van Wetenschappen | x | | x | | | | x | x | All |
| 2 | Nel [48] | 1927 | Geological Survey of South Africa | x | | | | | | x | x | All |
| 3 | Willemse [32] | 1937 | Transactions of the Geological Society of South Africa | x | x | x | | | | x | x | Holfontein (1), Kopjeskraal (7), El Dorado (8), Rensburgdrift (9) |
| 4 | Dietz [25] | 1961 | Journal of Geology | x | | | | | | | | ? |
| 5 | Hargraves [49] | 1970 | South African Journal of Geology | | | | | | x | x | x | Holfontein (1), Zandfontein (5), El Dorado (8), Rensburgdrift (9) |
| 6 | Wilshire [50] | 1971 | Journal of Geology | | x | x | | | | | x | ? |
| 7 | Bisschoff [16] | 1972 | Transactions of the Geological Society of South Africa | x | | x | | | | | | ? |
| 8 | French et al. [51] | 1989 | Proceedings of the 19th Lunar and Planetary Science Conference | | x | x | | x | | x | x | Holfontein (1), Kopjeskraal (7) |
| 9 | Reimold et al. [52] | 1990 | Proceedings of the 20th Lunar and Planetary Science Conference | x | | x | | | | x | x | Holfontein (1), Zandfontein (5), Kopjeskraal (7) |
| 10 | Walraven et al. [53] | 1990 | Tectonophysics | | | | x | | | | x | ? |
| 11 | Allsopp et al. [54] | 1991 | South African Journal of Science | | | | x | | | | x | ? |
| 12 | Bisschoff [55] | 1996 | South African Journal of Geology | | x | x | | | | | x | Rensburgdrift (9) |
| 13 | Kamo et al. [19] | 1996 | Earth and Planetary Science Letters | | | x | x | | | x | | Holfontein (1) |
| 14 | Koeberl et al. [56] | 1996 | Geology | | x | | | x | | | x | Kopjeskraal (7) |
| 15 | Therriault et al. [23] | 1996 | South African Journal of Geology | x | x | x | | | | x | x | All |
| 16 | Therriault et al. [7] | 1997 | South African Journal of Geology | | x | | | | | x | x | All |
| 17 | Henkel and Reimold [12] | 1998 | Tectonophysics | | | | | | x | | | ? |

**Table 1.** *Cont.*

| | Researchers | Year of Study | Publication | Field Studies | Bulk Geochemistry | Mineral Components | Geochronology | PGE and Isotope Geochemistry | Geophysics | Core Dikes | Core-Collar Dikes | Specific Dike(s) |
|---|---|---|---|---|---|---|---|---|---|---|---|---|
| 18 | Buchanan and Reimold [57] | 2002 | Meteoritics and Planetary Science | x | x | x | | | | | x | El Dorado (8), Rensburgdrift (9) |
| 19 | Henkel and Reimold [58] | 2002 | Journal of Applied Geophysics | | | | | | x | | | ? |
| 20 | Koeberl et al. [59] | 2002 | Geological Society of America Special1Paper | | x | | | x | | | x | Kopjeskraal (7) |
| 21 | Fagereng et al. [60] | 2007 | Contributions to Mineralogy and Petrology | x | | | | x | | x | | Holfontein (1) |
| 22 | Salminen et al. [61] | 2009 | Precambrian Research | | | | | | x | x | x | Holfontein (1), Zandfontein (5), Kopjeskraal (7), El Dorado (8), Rensburgdrift (9) |
| 23 | Moynier et al. [62] | 2009 | Earth and Planetary Science Letters | | | | | x | | | x | Kopjeskraal (7) |
| 24 | Lieger and Riller [63] | 2012 | Icarus | x | x | x | | | | | x | Kopjeskraal (7), El Dorado (8) |
| 25 | Wielicki et al. [64] | 2012 | Earth and Planetary Science Letters | | | x | x | | | | x | El Dorado (8) |
| 26 | Harris et al. [28] | 2013 | South African Journal of Geology | | | x | | x | | x | | Holfontein (1) |
| 27 | Huber et al. [65] | 2014 | Geology | | x | | | x | | | x | Kopjeskraal (7) |
| 28 | Wielicki and Harrison [66] | 2015 | Geological Society of America Special Paper | | | x | x | | | | x | El Dorado (8) |
| 29 | Reimold et al. [29] | 2017 | Geochemica et Cosmochimica Acta | | x | | | x | | x | x | Holfontein (1), Kopjeskraal (7) |
| 30 | Kovaleva et al. [67] | 2018 | South African Journal of Geology | | x | x | | | | x | | Daskop (4) |
| 31 | Kovaleva et al. [68] | 2018 | South African Journal of Geology | x | x | x | | | | x | | Holfontein (1), Daskop (4) |
| 32 | Fourie et al. [31] | 2019 | Meteoritics and Planetary Science | x | | | | | x | x | | Daskop (4) |
| 33 | Kovaleva et al. [69] | 2019 | Geology | | | x | | | | x | | Daskop (4) |

## 3. Results

### 3.1. Nature of Studies Found

Of the 33 studies that were identified, the earliest was the work of Hall and Mollengraff [24], and the most recent studies have taken place within the last year [31,69]. In the first 64 years of study on the granophyre dikes (1925–1988), there were seven peer-reviewed studies published (Figure 2). More recent publications came in two significant pulses: from 1989 to 2002, 13 studies have been published, and from 2008 to present, another 13 studies have been produced. Figure 2 shows the annual and decadal rate at which studies have been published. It is clear from Figure 2 that interest in the granophyre dikes increased greatly at the end of the 1980s, likely as a result of the identification of the Chicxulub impact structure [70,71] and the emerging technologies that allowed for a signature of the impactor to be tested. The studies were divided into the following categories of research: (1) fieldwork and mapping, (2) bulk geochemistry, (3) mineral components, (4) geochronology, (5) platinum group element (PGE) and isotope geochemistry, and (6) geophysical studies (Figure 3).

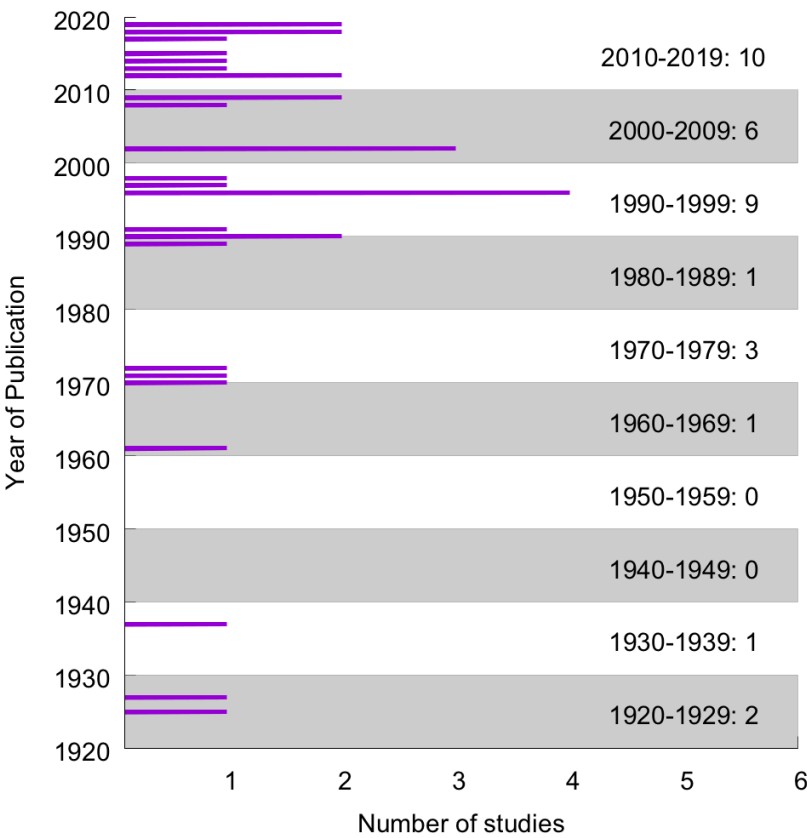

**Figure 2.** The number of published peer-reviewed studies per year and per decade.

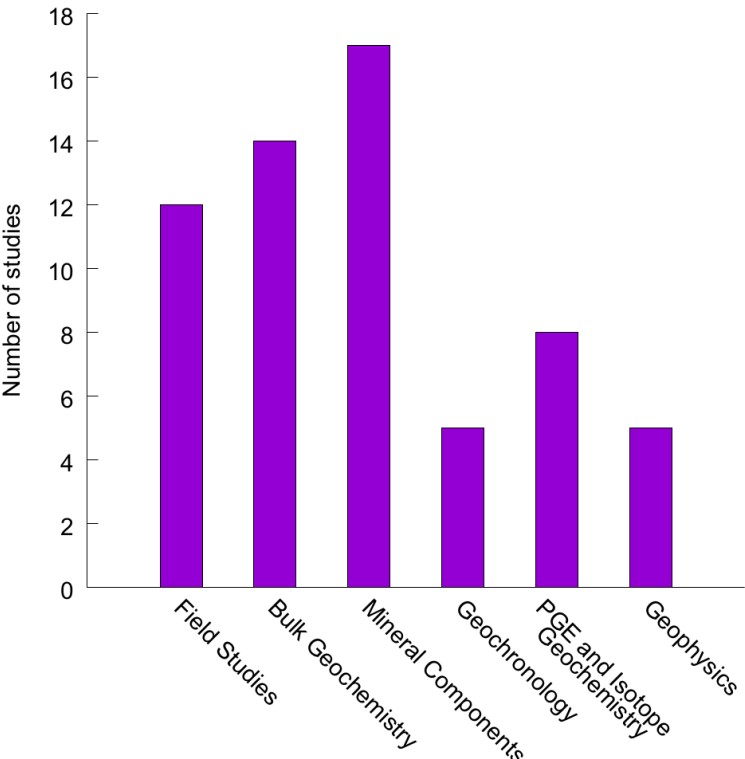

**Figure 3.** Number of published peer-reviewed studies organized by topic.

### 3.1.1. Fieldwork and Mapping

The initial studies of the Vredefort granophyre were the field and petrographic observations undertaken by Hall and Mollengraff [24], followed by geological mapping by Nel [48]. In these early works, remarkably insightful observations of the Vredefort structure were made. Already in 1925, it was well known that the Karoo volcano-sedimentary rocks and the underlying Transvaal basin possess a significant difference in depositional age. Hall and Mollengraff [24] stated that a "very powerful diastrophism" associated with a "point uplift" occurred to form the Vredefort structure. Amazingly, these authors [24] estimated an uplift of the granite in the core of the structure by at least 14 km, which is similar to the estimations made from numerical modeling 80 years later [14]. During these early field observations, the granophyre dikes were documented to cut across both the core and collar rocks [24,46]. Both studies argued that the emplacement of the dikes postdated uplift of the sedimentary strata, i.e., the granophyre cross-cuts impact-related faults and itself, is undeformed.

Starting from Dietz [25], the nature of published field studies has been to describe individual features of the granophyre dikes. Later, petrographic, and geochemical results of the investigations of the granophyre dikes were reported along with the field reports [32]. The most significant study that has investigated the field characteristics of the dikes was presented by Therriault et al. [23], who mapped all of the granophyre dikes in detail. In total, 12 studies have included a field component as a critical aspect of the study. These field investigations have shown that all of the dikes cross-cut their host rocks and are genetically unrelated to them. The dikes are variable in width, from a few meters to tens of meters. The core-collar dikes tend to be longer and wider than the core dikes (see below). Internal textures of the dikes have also been observed, including spherulitic textures [16] and pegmatitic veins within the dikes that have been interpreted as fluid escape structures [68]. Several workers have noted the variability of vol % of the lithic inclusions within the dikes [16,23,63,68]. However, the precise locations and distribution of the many internal features have not been well documented. Many of the dikes have not been described in detail, but are only described in broad terms along with the other dikes. In particular, the Mokwallo, Lesutoskraal, and Rietpoort dikes have not had their individual field characteristics described in peer-reviewed publications. Although field

analysis is the earliest technique employed to study granophyre dikes, there is still a need for further fieldwork to be undertaken.

### 3.1.2. Bulk Geochemistry

Bulk geochemical analysis of major and trace elements have been featured significantly in studies of the granophyre dikes, with 14 published papers that include a geochemical component. The most extensive geochemical study was undertaken by Therriault et al. [7,23], in which geochemical analyses of all nine granophyre dikes were presented. Both studies found that the composition of the granophyre dikes falls within a restricted, dacitic compositional range, with $SiO_2$ values between 65 and 68 wt %, $Al_2O_3$—between 12.0 and 13.0 wt %, $Fe_2O_3$—between 6.5 and 8.5 wt %, and $K_2O$, CaO, and $Na_2O$ values below 3.5 wt % (Table 2). These values have been reproduced both in the core and core-collar granophyre dikes by numerous studies. The results of bulk geochemical analyses from various workers have been used for modeling studies to suggest the lithologies contributed to granophyre (e.g., [72,73]), and for comparisons with other impact structures [30]. Therriault et al. [7] and Lieger and Riller [63] noted that there is a more mafic composition locally present within granophyre dikes in the core-collar boundary of the structure. Mafic portions were interpreted by Therriault et al. [7] as a result of an assimilation of epidiorite, and by Lieger and Riller [63] as a separate pulse of melt emplacement. To clarify this discrepancy, more field and geochemical investigations are needed.

**Table 2.** Mean major element composition of granophyre dikes. Data from [7,24,32,52,63], n = 60.

| Oxide | Average Composition |
|---|---|
| $SiO_2$ | 66.72 ± 1.12 |
| $TiO_2$ | 0.49 ± 0.09 |
| $Al_2O_3$ | 12.64 ± 0.34 |
| $Fe_2O_3$ | 7.38 ± 0.58 |
| MnO | 0.14 ± 0.03 |
| MgO | 3.6 ± 0.29 |
| CaO | 4.65 ± 4.51 |
| $Na_2O$ | 2.63 ± 0.42 |
| $K_2O$ | 2.18 ± 0.24 |
| $P_2O_5$ | 0.13 ± 0.06 |

### 3.1.3. Petrography and Mineral Components

Petrographic observations of the mineral phases within the granophyre have been a focus of studies of the granophyre dikes from the earliest to the most recent papers. These studies have been used to characterize the nature of the granophyre melt rock, as well as analyze lithic inclusions within the granophyre dikes. In total, 17 papers have included a significant petrographic component. Hall and Mollengraff undertook the earliest such study [24], and the most recent studies have also included a petrography component [67,68]. The findings of these studies are broadly consistent with one another, describing elongate orthopyroxene crystals and minor clinopyroxene enclosed within a micropegmatitic groundmass of feldspar and quartz. The various textures present in the different granophyre dikes were first shown by Bisschoff [16], who characterized three types of granophyre: (1) A fine-grained granular rock that is the principal type in all examined dikes. (2) A spherulitic type in which the spherulites are up to 2–3 cm in diameter and are formed on the marginal facies of some dikes. (3) A spherulitic type with small spherulites about 0.5–1.0 cm in diameter that is found in the chilled margins of the dikes. In his work, the pyroxene in the granophyre was petrographically determined to be bronzite based on the optical axis orientation [16], so that for many years the granophyre was referred to as the "bronzite granophyre," or as "enstatite granophyre." Later analysis [23,52,68] demonstrated a hypersthene composition of the pyroxene. The textural observations of Bisschoff, however, have been confirmed through the work of others [23]. Linear structures within the core granophyre dikes were petrographically characterized as coarse-crystalline pegmatoidal veins with minerals compositionally

indistinguishable from those in the surrounding groundmass [68]. Hydrous phases have not been observed in great abundance, but biotite, chlorite, apatite, amphibole, and various spinels have been documented locally in small concentrations [23].

There have been five studies that generated electron microprobe analysis (EMPA) data of the major mineral phases within the granophyre [16,23,52,57,68]. Chemical maps generated by EMPA have also been presented [68]. The results of the chemical analysis of pyroxenes and feldspars are presented in Figure 4. The chemistry of the pyroxenes is primarily hypersthene in the core dikes [68], but pigeonite and augite compositions are present in the core-collar dikes [23]. The alkali feldspar compositions range from sanidine to microcline, and the plagioclase compositions are predominantly labradorite [23,68].

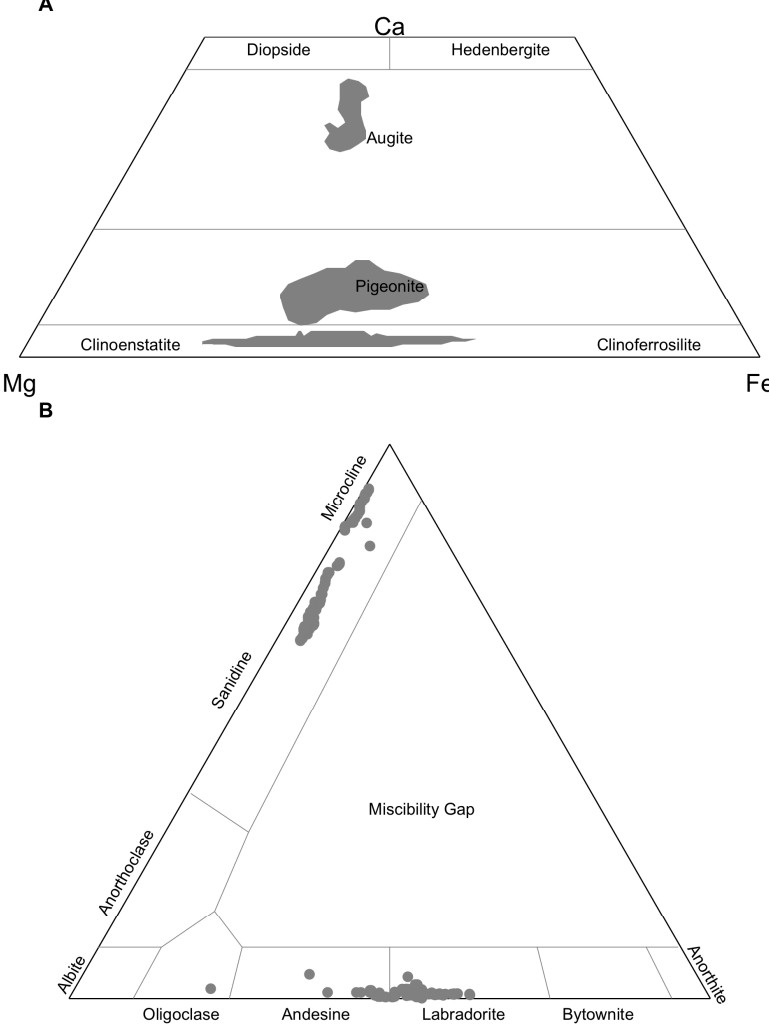

**Figure 4.** Pyroxene (**A**) and feldspar (**B**) mineral chemistry from electron microprobe analyses. Data from [23,68].

In addition to the studies of the granophyre itself, several studies have focused entirely on the inclusions within the granophyre. The inclusions were shown to preserve shock features by French et al. [51] and Reimold et al. [52], in contrast with the typical surficial rocks in the Vredefort structure, where shock features are mostly recrystallized [74]. Buchanan and Reimold [57] reported shocked quartz, decomposed feldspar, melt pockets, and partial melting in 30 quartzite and granitoid inclusions extracted from the granophyre. The further development of such studies was strongly encouraged by French and Nielsen [72], and the relevance of these investigations was recently demonstrated by Kovaleva et al. [67], who discovered a granite fragment with an impact-generated

pseudotachylite vein enclosed within the granophyre and provided a detailed examination of the associated shock features. Later, shocked zircons from the same clast were studied in detail [69].

Shocked zircon was first identified and characterized at the Vredefort structure by Kamo et al. [19]. In particular, the heavy mineral separate from granophyre revealed the first zircon with granular morphology from the Vredefort impact structure. Dating of the granular zircon indicated possible Pb loss at ca. 2.0 Ga and ca. 1.0 Ga. Unshocked zircons, extracted from granophyre, indicated the age of Archean crystallization [19]. After this study was undertaken, the electron backscatter diffraction (EBSD) technique was adopted by the geoscience community [75] and later began to be utilized as a tool for understanding shocked accessory minerals [76,77]. Numerous EBSD analyses of shocked zircons that were eroded from the Vredefort structure and sampled in detrital settings have been done (e.g., [78]). Because these papers report zircons that are removed from their original context, it is unclear which impactites have hosted the analyzed zircons. Some of the detrital zircons were likely derived from the rock fragments enclosed within impact melt, but this cannot be conclusively demonstrated.

There have been three studies that analyzed in situ zircons from the Vredefort granophyre. The first [64] examined 13 zircons from the Vredefort granophyre, but based on the Ti-in-zircon thermometer, the zircons were determined to be inherited preimpact. The second study [66] further examined the same suite of zircons and compared the formation of the zircon in impact melt to lunar zircons, with a particular emphasis on using zircons in impact melt for dating an impact event. Both studies attempted to determine the ages of zircons within the granophyre (Figure 5) but showed that the grains were likely inherited Archean zircons rather than newly grown, and experienced a significant loss of radiogenic elements resulting from heating and partial melting. The study of shocked zircons in a clast in granophyre by Kovaleva et al. [69] has produced EBSD maps of the granular zircons in situ. In this work, the origin of the clast with the shocked granular zircon was demonstrated to be in the upper portions of the impact structure, suggesting that the clast had traveled up to 10 km downwards within the granophyre. The need for further analysis of such lithic clasts within the granophyre is a major area where future work can be focused [69,72].

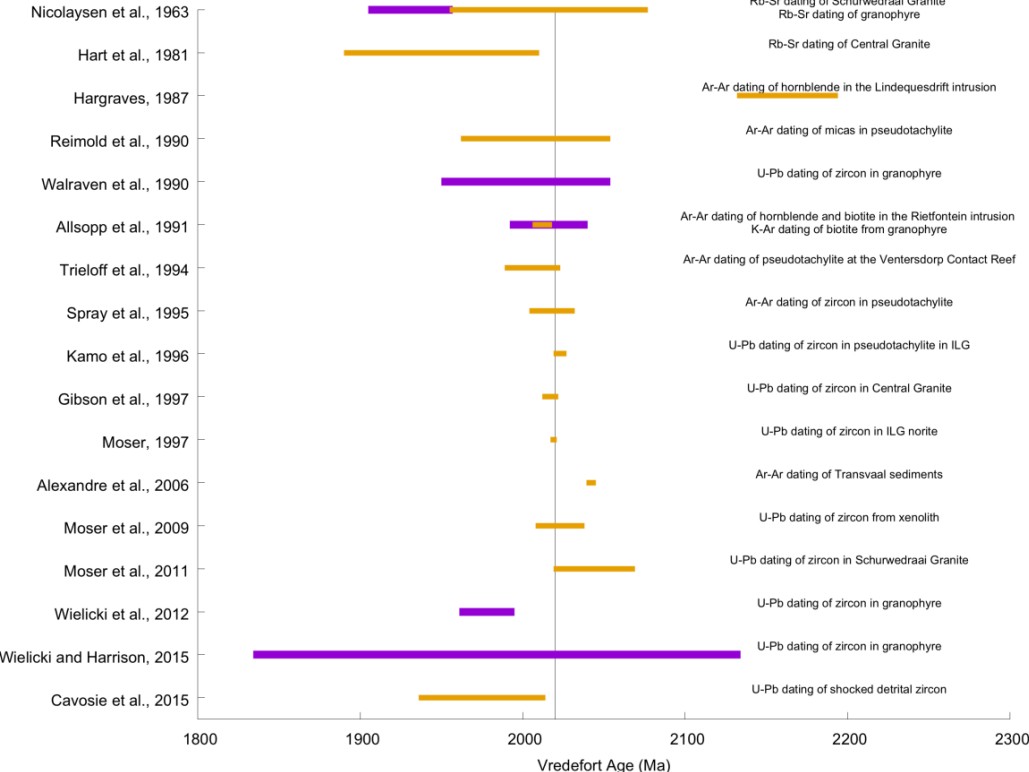

**Figure 5.** Attempts to determine the age of the Vredefort impact structure. Purple lines represent ages derived from the granophyre, while orange lines represent ages derived from other impactites. At least five studies have determined ages from the granophyre dikes. The lengths of the lines represent the 2σ age range presented by the workers. The black vertical line is plotted at 2020 Ma. Note that ages obtained after 1997 have been attempts to constrain the effects of impact deformation rather than to improve the age accuracy of the impact event. Additionally, [19] produced ages of zircons extracted from granophyre that have discordant ages and do not plot on this figure. The generally accepted age of the Vredefort event is considered to be 2020 ± 2 Ma. Data from [19,20,52–54,64,66,78–87]. ILG = Inlandsee Leucogranofels.

### 3.1.4. Geochronology and Age Determination

The Vredefort granophyre has been recognized as occurring late in the development of the Vredefort structure by even the earliest workers [24]. Therefore, it was the first target of an attempt to date the impact structure [79], although this first age was only reported in an abstract and not a peer-reviewed publication (Figure 5). Subsequently, the granophyre was successfully dated by four peer-reviewed papers. Still, the most precise ages of the Vredefort event have been obtained from the Central Anatectic Granite so that by 1997, the generally agreed-upon age of the Vredefort structure has been considered to be 2020 ± 2 Ma (e.g., [41,80]).

Numerous studies, not all of which have been peer-reviewed, have acquired the age of the Vredefort impact event. The ages have been acquired from various rock types, including the granophyre itself, which postdates the impact event, the Central Anatectic Granite, which also postdates the impact event, pseudotachylites, which are probably coincident with the impact event, and the hydrothermally altered Witwatersrand basin, in which the postimpact hydrothermal system may have been active for a significant period of time following the impact event. The ages related to the Vredefort structure, along with the method and target for dating, are presented in Figure 5. Notably, the earliest acquired age of the Vredefort structure in 1963 was accurate. The ages of zircons extracted from the granophyre tend to have less precise dates than those obtained via other methods, which is explained to be due to the shock characteristics and partial resetting of zircons within the granophyre [64]. For instance, Kamo et al. [55] produced significantly discordant ages and concordant Archaean ages of zircons extracted from granophyre but did not argue that these ages date the structure. The imprecise ages derived in the 2000s and 2010s were not attempting to date the impact structure, but rather utilize the known age of the Vredefort structure to examine other questions, such as the nature of resetting of ages in shocked zircon (e.g., [78]).

### 3.1.5. Platinum Group Element and Isotope Geochemistry

Starting with French et al. [51], there have been five studies that have used geochemical techniques designed to search for a signature of the impactor within the impact melt, including examination of PGEs and isotopic systems (a summary of such techniques can be found in [88]). The first study examined the Ir content of the granophyre dikes and compared these values to surrounding host rocks, determined that there was not an impactor component present [51]. A later Re-Os analysis (using a subset of samples previously analyzed by [52]) indicated that the granophyre dikes include a small (ca. 0.2 wt %) chondritic component, but only considered the Ventersdorp lavas and Witwatersrand shale in comparison to the granophyre Re-Os data [56]. Further work analyzed the Cr isotope compositions of the same suite of samples as [56], as well as the Ir and Pt content of these samples, but the results were indistinguishable from a purely terrestrial signal [59]. Additionally, the W isotope composition of the same suite of samples was analyzed [62], and these analyses did not identify a clear meteoritic component either. Finally, [65] reported platinum group element (PGE) analysis of the same suite of samples as earlier studies [56,59,62], and compared these results to a likely occurrence of ejecta from the Vredefort impact, concluding that the PGE patterns were similar. Therefore, although four separate techniques analyzed the same set of samples, the only technique that has suggested a

presence of the impactor signature within the granophyre dikes is the Re-Os analysis [56]. It is not clear that a chondritic contribution is the only possible explanation for the Re-Os isotopic data, as a komatiitic component could potentially explain the same trends [89], and komatiites are known to be in the Vredefort target rocks. Additionally, a later conference abstract suggested that the Re-Os contents of the Kopjeskraal dike did not show the same trend (data were not included in the abstract) [90]. To clarify the matter, further isotopic analyses of the granophyre are recommended.

Apart from the studies that have used trace element isotopes to search for the impactor component, isotopic characterization of the granophyre dikes has been the focus of three studies. The first such study was undertaken by Fagering et al. [60], who reported $\delta^{18}$O and $\delta$D data from a variety of rocks in the Vredefort structure, including two samples from the Holfontein granophyre. They demonstrated that the $\delta^{18}$O value of the granophyre is distinct from the pseudotachylites and other rocks in the structure. A later study expanded the dataset with two additional granophyre samples and a wider array of basement samples [28]. The findings of both Fagering et al. [60] and Harris et al. [28] were that the $\delta^{18}$O composition of the granophyre is remarkably consistent (7.6‰). Both studies focused on the Holfontein dike and did not compare the results to other dikes in the core or the core-collar boundary. In a separate study, the $^{87}$Sr/$^{86}$Sr, $^{87}$Rb/$^{86}$Sr, $^{143}$Nd/$^{144}$Nd, and $^{147}$Sm/$^{144}$Nd values, as well as the U decay series of Pb isotopes, were reported from two samples taken from the Holfontein dike and three samples from the Kopjeskraal dike, as a means of tracing the provenance of the granophyre and pseudotachylites [29]. The study primarily concluded that pseudotachylites do not require a granophyre component to explain their composition and that more isotopic studies on the melt rocks are warranted. Therefore, there have been a total of six samples from the Holfontein dike and three samples from the Kopjeskraal dike that have been analyzed regarding isotope geochemistry.

### 3.1.6. Geophysics

A variety of geophysical techniques have been undertaken at the Vredefort structure, with the majority being regional studies focusing on the impact structure as a whole. Five studies have applied geophysical methods specifically to the granophyre dikes. The earliest such work was conducted in 1970 by Hargraves [49], who conducted a paleomagnetic study of the granophyre dikes. The study was used to constrain the paleogeographic location of the Kaapvaal craton and to support the impact hypothesis. The next geophysical study that explicitly produced data relating to the granophyre dikes was only in 1998 when Henkel and Reimold reported a density of 2.787 mg/m$^3$ and magnetic susceptibility of 0.01 SI for the granophyre [12]. Later, the same authors analyzed an additional 25 granophyre samples to update their result to 0.004–0.3 SI [58]. A paleomagnetic study and report of the magnetic properties of 14 samples from granophyre dikes and 15 samples of pseudotachylite were performed by Salminen et al. [61]. Among these analyses, 8 of the 14 samples had high Q values and anomalous directions of magnetization, but the remaining samples yielded a paleolatitude of 25.1° and a paleolongitude of 43.5°. The magnetic and electrical resistivity survey of one of the core dikes found that the signal of the highly resistive material, corresponding to the granophyre, disappears <5 m below the present-day surface, which is interpreted to show the shallow penetration depth of the dike [31]. This result indicates a strong need for further geophysical studies on the granophyre dikes.

### 3.2. Studies on Individual Granophyre Dikes

The granophyre dikes are divided into the core and core-collar dikes. Textural and dimensional differences between these dikes have been observed, with the core-collar dikes being wider and longer than the core dikes, but having the same geochemical composition [16]. Core-collar dikes are mostly granular, while the core dikes contain spherulitic and pegmatiodal textures more often. Pyroxenes in the core-collar dikes have higher Ca content than the core dikes (Figure 4). There has been an unequal number of papers, with 22 studies including core-collar dikes as part of their analysis and 17 studies including core dikes (eight studies included both; Figure 6 and Table 3). Six studies have analyzed

granophyre samples without making clear which dike was analyzed. For the locations of the dikes, see Figure 1 or the detailed descriptions in Therriault et al. [23].

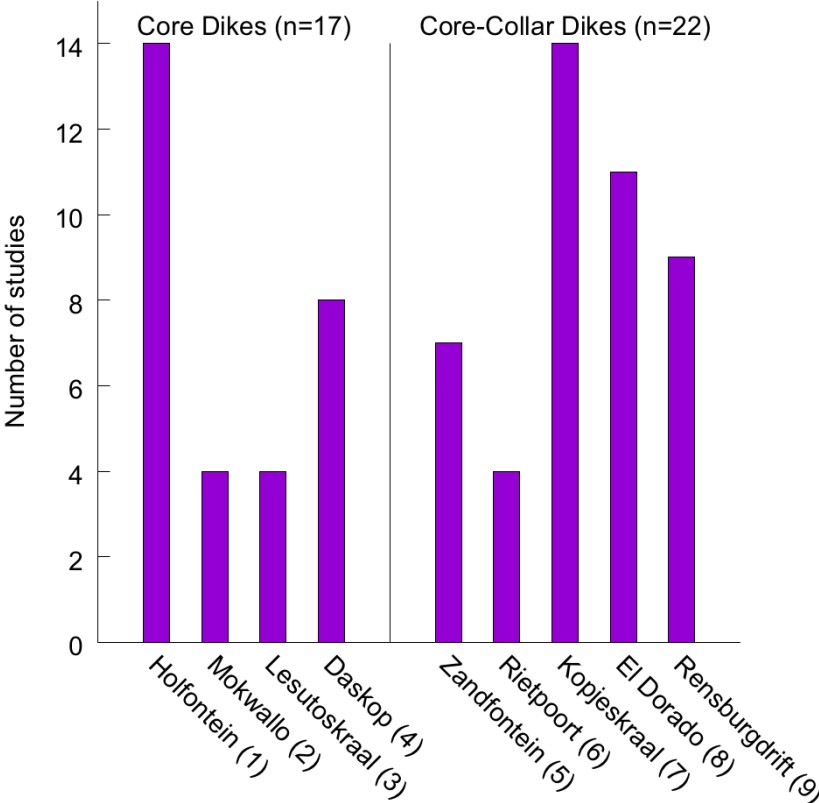

**Figure 6.** Number of published peer-reviewed studies per individual granophyre dike.

### 3.2.1. Core-Collar Dike Studies

At least 22 distinct scientific works have studied the core-collar dikes. The Kopjeskraal dike has been the target of the majority of these studies, with 14 studies on this dike in particular. The El Dorado and Rensburgdrift dikes, which are located on properties adjacent to Kopjeskraal, have also been the target of a large number of studies, with 11 and 9 studies, respectively. The dikes in the North of the structure, the Rietpoort, and Zandfontein dikes, have been studied less, with only four studies concerning Rietpoort dike (with none being dedicated to this dike only), and seven studies on the Zandfontein dike.

### 3.2.2. Core Dike Studies

At least 17 scientific works have studied the core dikes. The most studied core dike is, by far, the Holfontein dike, with 14 distinct studies. Before 2018, none of the other core dikes has been the focus of any study, with each of them being surveyed only in papers that investigated all of the dikes in the structure simultaneously. Since then, four studies that include analysis of the Daskop dike have been published, with one study being a nondestructive geophysical survey [31] and one including nondestructive field observations of the Daskop dike (together with the petrography and geochemistry of the Holfontein dike) [68]. The other two papers [67,69] investigated a unique sample that was extracted from the dike with the permission of the South African Heritage Resource Agency [91]. In general, studies of the Daskop dike have been limited because of the presence of preindustrial petroglyphs.

**Table 3.** Studies performed on each granophyre dike and accessibility of each dike. Many studies include multiple methods. Numbers in brackets after the dike names indicate the numbering used by [23].

| | Core | | | | Core-Collar Boundary | | | | | |
|---|---|---|---|---|---|---|---|---|---|---|
| | **Holfontein (1)** | **Mokwallo (2)** | **Lesutoskraal (3)** | **Daskop (4)** | **Zandfontein (5)** | **Rietpoort (6)** | **Kopjeskraal (7)** | **El Dorado (8)** | **Rensburgdrift (9)** | **Unknown** |
| Total studies performed | 14 | 4 | 4 | 8 | 7 | 4 | 14 | 11 | 9 | 6 |
| Field Studies | 9 | 3 | 3 | 5 | 4 | 3 | 6 | 6 | 5 | 2 |
| Bulk Geochemistry | 7 | 2 | 2 | 4 | 2 | 2 | 9 | 5 | 5 | 1 |
| Mineral Components | 10 | 2 | 2 | 5 | 3 | 2 | 6 | 7 | 5 | 2 |
| Geochronology | 3 | 0 | 0 | 0 | 0 | 0 | 0 | 2 | 0 | 2 |
| PGE and Isotope Geochemistry | 4 | 0 | 0 | 0 | 0 | 0 | 6 | 0 | 0 | 0 |
| Geophysics | 4 | 0 | 0 | 1 | 2 | 0 | 1 | 2 | 2 | 2 |
| Accessibility | Public Land | Public Land | Private game reserve | Private farm | Private farm | Private farm | Private farm | Private game reserve | Private farm | - |

## 4. Discussion

Despite nearly 100 years of study of the Vredefort impact structure, only a small number of published academic studies have produced data regarding the granophyre dikes; on average, there have been only slightly more than one publication every three years (0.35 studies per year from 1925 to present). The number of studies has increased in the last three decades, with 25 publications in the last 30 years, or an average of five studies every six years. By contrast, a Science Direct search for "Sudbury Offset Dike" reveals that 39 peer-reviewed manuscripts have been published concerning the Sudbury Offset Dikes in just the last five years (7.8 studies per year).

The types of research on the granophyre dikes have changed over time. The early work on the granophyre focused on the basic properties of the dikes, such as field mapping, petrography, and bulk geochemistry. This work gave way to the attempts to determine the age of the Vredefort structure, which was a major point of discussion up until the late 1990s. A somewhat longer and ongoing discussion has taken place attempting to determine if an impactor component was present in the dikes, with the most recent contribution in 2014. Meanwhile, geophysical studies have continued to be undertaken from 1970 up until the present. Shock deformation features in zircon from granophyre were first discovered in 1996, but with recent advances in the study of accessory minerals affected by the impact event, there have been three additional studies in the 2010s that have investigated zircon from granophyre. In the late 2000s, the first peer-reviewed stable isotope study was published, and stable isotopes have continued to be investigated until the present. These changing trends in research reflect the state of understanding at Vredefort, the interests of the larger scientific community, and also the technological capabilities of researchers.

The research on the granophyre dikes has been driven in large part by the accessibility of particular sites. In the collar of the Vredefort structure, the Kopjeskraal granophyre dike has been the most studied. A significant reason for such attention that this dike has received is the openness of the current landowner to scientific investigations. At least five universities in South Africa take undergraduate students to the Kopjeskraal farm for annual field trips. In the core of the structure, the Holfontein dike has been the most studied, because it is on public land that does not require permission to access and sampling. Besides this, Holfontein dike lies outside the boundaries of the proposed UNESCO World Heritage site. The frequency with which this dike has been visited is obvious from the numerous hammer- and saw-marks present on the dike. By contrast, the Lesutoskraal dike has not been well studied, owing in large part to the fact that this dike lies within the buffer zone of the UNESCO World Heritage site and is located on a private game preserve that is not easily accessible. The core-collar dikes have generally received more attention than the core dikes, which is likely because the collar is easier to access and present broader outcrops.

An additional reason for the focus of studies on particular dikes is that sample suites have been re-analyzed numerous times. Numerous sets of samples have been analyzed in two or more studies, with one sample set having been analyzed in at least five studies [52,56,59,62,65]. Although it is good practice to obtain the maximum possible data from unique samples once they are collected, this demonstrates that the number of unique samples is lower than the number of individual studies on dikes. Greater diversity in samples is needed for the Vredefort granophyre to be fully understood.

Many research questions concerning the granophyre dikes remain to be explored. The detailed distinctiveness between the dikes has not been evaluated, and very few studies have focused on fully characterizing particular dikes. Previous workers have established the general compositional similarity between the granophyre dikes in the core and the core-collar boundary. Still, the anomalous mafic phase identified by [7] and [63] has not been thoroughly investigated. Despite the great abundance and variety of inclusions that are present in the granophyre, few such inclusions have been investigated. Inclusions that have been studied (i.e., [57,67,69]), are remarkably significant for new insights into the impact process.

In the same way, while in situ studies of shocked zircon from the granophyre dikes have not been thoroughly undertaken, other accessory minerals have not been investigated at all for potential

shock deformation. Questions concerning the impactor component remain, as there have only been a small handful of studies that investigated this, with ultimately only one set of samples from one dike having been investigated. Only one study has suggested a quantifiable concentration of the impactor component. Geophysical analysis of the granophyre dikes has already demonstrated that there are anomalies in both the host rocks around the granophyre dikes and with the limited penetration depth of the dikes themselves [31]. More such studies are needed to understand these anomalies.

Despite the common assertion that everything has been done and known concerning the Vredefort impact structure, the reality is that many aspects have not been well studied. The Vredefort granophyre is an exceptionally important component of the structure that is deserving of considerable attention in order to understand the formation and development of the structure.

**Funding:** This work is produced within the framework of interdisciplinary project GRAVITAS ("Geological Research and Analysis of Vredefort Impact with Timely Anthropological Studies"), a grant received from the Directorate for Research Development, University of the Free State, 2019–2020.

**Acknowledgments:** The authors thank Martin Clark for assistance with generating Figure 1. The authors acknowledge the helpful and insightful comments of the three reviewers.

**Conflicts of Interest:** The authors declare no conflicts of interest.

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
