# Peer review of "Identifying Gaps in the Investigation of the Vredefort Granophyre Dikes: A Systematic Literature Review"

_geosciences, doi:10.3390/geosciences10080306_

Round 1
Reviewer 1 Report
Review of the manuscript entitled: Identifying gaps in the understanding of the Vredefort granophyre dikes: A systematic literature review
by Huber M.S. and Kovaleva E.
The paper focused on review of the total number of unique peer-reviewed scientific works in English that have published the results of study of the Vredefort granophyre dykes. The importance of the Vredefort granophyre dykes is emphasized by the fact that the impact melt sheet has been completely erosionally removed, the granophyre is its only remnant available for study. Obviously, this review can be useful for understanding the formation of other basin-sized impact structures on Earth, Moon and Mars, especially when considering significantly eroded impact craters. The publication of the review based on the results of a hundred-year history of the studies of granophyre dikes of a unique basin-sized crater is now completely timely. The authors rightly highlight an increase in interest to the research of granophyre dikes at last decade, which is associated both with advances in understanding their genesis and technological capabilities of researchers. Certainly review will initialize the emergence of new research papers.
Dividing the 33 studies into 7 categories of research (fieldwork and mapping, petrography and shocked mineral studies, major and trace element geochemistry, dating the impact, search for impactor component, other isotopic studies, geophysical studies) is original and reasonable. It allows better orientate in the literature and rapidly enter into the subject of studying the Vrederfort granophyre dikes. The gaps in studies and selected possible directions for further investigations are well shown in review. Geophysical researches involved in the review paper emphasizes the importance of these applying along with direct study of rocks by geologists and mineralogists. In my opinion, the most interesting are the generalization data on petrography and shocked mineral studies, the comparison of granophyre dikes with pseudotachylites and the discussion of the formation and development of basin-sized Vredefort impact structure.
Although authors did a good job with their work, but minor changes in the manuscript is needed. References need to be carefully checked for typos and correct formatting.
Some detailed comments:
Line 116 – The work could be more objective when considering publication not only in English. Perhaps the authors have information about their existence or absence. Please mention in the text the most important, if any.
Line 138, Table title – Please use in table title “granophyre dykes”.
Line 138, Table, Last column – Indicate in the table what the number in brackets after the name of the dyke means.
Line 138, Table – Could you please use references in column "researchers" or others. It will significantly help readers.
Line 144 – Not 12, but 13 articles.
Line 153, Fig.2 – The number of publications for all years is 34, not 33 (the sum of the values in brackets). For “1960-1969”, the number 2 was written, but should be 1.
Line 155, Fig.3 – The number of studies in “Stable isotopes” not 4, but 3.
Line 178 – Not 15 papers, but 15.
Line 232 – The figure shows 17 publications, and the text mentions 16. Please correct.
Line 242, Fig.4. – References are not correct. Please fix «Allsopp et al., 1990» on «Allsopp et al., 1991». The dating of [Nicolaysen et al., 1963] are shown in the figure, but the article is not in the table 1. Please correct.
Line 496 – References is not correct. “2015” is lost in [54] reference.
Line 516 – References is not correct. Please fix «1990» on «1991» in [62] reference.
Author Response
Our responses below in red.
The paper focused on review of the total number of unique peer-reviewed scientific works in English that have published the results of study of the Vredefort granophyre dykes. The importance of the Vredefort granophyre dykes is emphasized by the fact that the impact melt sheet has been completely erosionally removed, the granophyre is its only remnant available for study. Obviously, this review can be useful for understanding the formation of other basin-sized impact structures on Earth, Moon and Mars, especially when considering significantly eroded impact craters. The publication of the review based on the results of a hundred-year history of the studies of granophyre dikes of a unique basin-sized crater is now completely timely. The authors rightly highlight an increase in interest to the research of granophyre dikes at last decade, which is associated both with advances in understanding their genesis and technological capabilities of researchers. Certainly review will initialize the emergence of new research papers.
Dividing the 33 studies into 7 categories of research (fieldwork and mapping, petrography and shocked mineral studies, major and trace element geochemistry, dating the impact, search for impactor component, other isotopic studies, geophysical studies) is original and reasonable. It allows better orientate in the literature and rapidly enter into the subject of studying the Vrederfort granophyre dikes. The gaps in studies and selected possible directions for further investigations are well shown in review. Geophysical researches involved in the review paper emphasizes the importance of these applying along with direct study of rocks by geologists and mineralogists. In my opinion, the most interesting are the generalization data on petrography and shocked mineral studies, the comparison of granophyre dikes with pseudotachylites and the discussion of the formation and development of basin-sized Vredefort impact structure.
Although authors did a good job with their work, but minor changes in the manuscript is needed. References need to be carefully checked for typos and correct formatting.
Some detailed comments:
Line 116 – The work could be more objective when considering publication not only in English. Perhaps the authors have information about their existence or absence. Please mention in the text the most important, if any.
We did not come across many works at all that were not in English, and certainly no non-English works have been cited concerning the Granophyre. We have mentioned the lack of such works in the text.
Line 138, Table title – Please use in table title “granophyre dykes”.
Fixed
Line 138, Table, Last column – Indicate in the table what the number in brackets after the name of the dyke means.
Fixed.
Line 138, Table – Could you please use references in column "researchers" or others. It will significantly help readers.
Added
Line 144 – Not 12, but 13 articles.
Fixed.
Line 153, Fig.2 – The number of publications for all years is 34, not 33 (the sum of the values in brackets). For “1960-1969”, the number 2 was written, but should be 1.
The total number of publications is 33. We have fixed Figure 2.
Line 155, Fig.3 – The number of studies in “Stable isotopes” not 4, but 3.
Fixed.
Line 178 – Not 15 papers, but 15.
Fixed
Line 232 – The figure shows 17 publications, and the text mentions 16. Please correct.
Fixed
Line 242, Fig.4. – References are not correct. Please fix «Allsopp et al., 1990» on «Allsopp et al., 1991».
Corrected
The dating of [Nicolaysen et al., 1963] are shown in the figure, but the article is not in the table 1. Please correct.
Nicolaysen et al., 1963, is an abstract and not a peer-reviewed publication, so it is not included in Table 1. It was included, despite the methodology of our study, as it is the very first – and accurate – dating attempt of the structure and thus is remarkable. We have included wording in the relevant section to reflect this.
Line 496 – References is not correct. “2015” is lost in [54] reference.
Corrected
Line 516 – References is not correct. Please fix «1990» on «1991» in [62] reference.
Fixed
Reviewer 2 Report
Dear Matthew and Elizaveta,
I have reviewed your manuscript: “Identifying gaps in the understanding of the Vredefort granophyre dikes: A systematic literature review”.
I enjoyed reading the manuscript and agree that a review of the Vredefort Granophyre would be an important contribution to the literature. Nevertheless, I believe that the manuscript requires major revision before publication. The content that is currently in the manuscript is generally acceptable for publication, however, as a review, I think it is missing an important element. A review paper should critically evaluate the literature to provide a summary of the state of knowledge on a subject. I came away from reading your manuscript without being able to answer some simple geological questions about what the granophyre is, instead, I felt that I had learnt about the papers that describe the granophyre. I think that by adding a more critical evaluation and summary of the state of knowledge on the Vredefort Granophyre, stating what the Granophyre is and what it tells us about the bigger picture, the paper will be more widely citeable. Without it, your paper runs the risk of being used as a reading list, without ever being cited itself.
Attached, you will find a comprehensive list of the general and specific comments that I feel should be addressed before publication. I hope that you find them to be useful for the improvement of the manuscript.

Author Response
Our responses in the attached document in Red.

Reviewer 3 Report
Review for Geoscience #863323:
Identifying gaps in the understanding of the Vredefort granophyre dikes: A systematic literature review
The paper presents a literature overview on the granophyre dikes of the Vredefort impact structure. The text is well-written, and the literature overview in Table 1 is useful to a geoscientific audience. It is stated that the purpose of the literature study is to identify gaps in the knowledge of the Vredefort granophyre dikes, but the authors do not explore or identify such gaps in the paper to great depth. The major conclusion of the paper is stated in the abstract as “[…] a relatively small number of studies on this important rock type, and more studies are necessary for the granophyre dikes to be truly understood. “. The authors may be correct with this conclusion, but in my opinion, they do not provide convincing evidence for it in the manuscript (see e.g. my point 1 below). Together with some other critical aspects that I found in the paper that are listed below, I therefore recommend rejection of the paper, with the encouragement to the authors to resubmit it. Please find below some comments on the paper / advise to the authors that are needed to the paper before acceptance:
1) The authors conclude that „Despite nearly 100 years of study of the Vredefort impact structure, a surprisingly small number of published academic studies have produced data regarding the granophyre dikes“; because,
„on average, there have been only slightly more than one publication every three years (0.35 studies per year from 1925 to present). The number of studies has increased in the last three decades, with 25 publications in the last 30 years, or an average of five studies every six years.“
I do not see the causality in this statement. I think that 25 publications in 30 years on a rock type as specific as the Vredefort granophyre dikes is quite a lot. The authors can convince the reader otherwise by comparing their literature compilation with a literature compilation of e.g. the melt sheet of the Sudbury structure. If there would be 2500 publications on the Sudbury impact melt from the past 30 years, then the authors are correct that the Vredefort granophyre dikes are underrepresented in the academic literature. Without making such a comparison and without putting their literature study into perspective, it cannot be concluded that a “surprisingly small number” of published academic studies are available on the Vredefort granophyre dikes.
In general, I also advice to avoid words such as “surprisingly” from statements like this. “Surprisingly” in this particular context is misleading to the readers, and it has no scientific value.
2) Table 1 - The division of studies by topic as well as by methods of investigation is somewhat random. In the current version of the manuscript, analytical methods (i.e., scientific results) such as “stable isotopes“ are presented along with research topics such as “impact component” (i.e., an interpretation of scientific data). The authors can maintain the division of topics as they like, but my advice is to more clearly distinguish between analytical methods and data on one hand (geochemistry, geophysics etc) and focal points of previous studies (impactor component) on the other hand, possibly be restructuring the table.
3) The section on isotope geochemistry needs a careful iteration! Several sentences in the current version of this section are semantically and sometimes scientifically incorrect. Some suggestions:
- Line 254: Cr isotopic content --> please change to Cr isotope compositions
- Line 256: W isotopic content --> please change to W isotope compositions
- Line 264: δ18O and δD isotopic data --> remove „isotopic“
- (...) and the δ18O in more detail. --> I don’t understand this part of the sentence, please clarify.
- „Later work reported 87Rb, 86,87Sr, 147Sm, 143,144Nd, using isotopic ratios to constrain the possible assimilated components within the granophyre [27]“. --> Please carefully check this sentence, because it contains several incorrect statements. Previous work would have reported 86Sr/88Sr and 143Nd/144Nd ratios. These are the measured values and should be reported as ratios. Note also that 88Sr is in the denominator of the Sr isotope ratio, and not 87 In contrast, 87Rb and 147Sm are not measured values but Rb/Sr and Sm/Nd ratios are reported based on Rb and Sr concentrations, as measured by isotope dilution. Also, the focus of the studies that are referred to would have been variations in radiogenic isotope abundances (86Sr, 144Nd), and variations related to stable isotope fractionation. I would generally merge this information with the section on major and trace element geochemistry (isotopes were mainly used as tracers for provenance and/or geochemical mixing processes).
4) The authors present a thorough literature compilation and summarize the results of previous publications, but the overall discussion of these previous results is really thin. It is pointed out in the manuscript that the latest review paper on the Vredefort granophyre dikes did not include the bulk of the published data. I would recommend to include a more thorough scientific discussion of the literature - this is a big chance for the authors to increase the impact of the paper.
Author Response
Our responses below in red.
The paper presents a literature overview on the granophyre dikes of the Vredefort impact structure. The text is well-written, and the literature overview in Table 1 is useful to a geoscientific audience. It is stated that the purpose of the literature study is to identify gaps in the knowledge of the Vredefort granophyre dikes, but the authors do not explore or identify such gaps in the paper to great depth. The major conclusion of the paper is stated in the abstract as “[…] a relatively small number of studies on this important rock type, and more studies are necessary for the granophyre dikes to be truly understood. “. The authors may be correct with this conclusion, but in my opinion, they do not provide convincing evidence for it in the manuscript (see e.g. my point 1 below). Together with some other critical aspects that I found in the paper that are listed below, I therefore recommend rejection of the paper, with the encouragement to the authors to resubmit it. Please find below some comments on the paper / advise to the authors that are needed to the paper before acceptance:
1) The authors conclude that „Despite nearly 100 years of study of the Vredefort impact structure, a surprisingly small number of published academic studies have produced data regarding the granophyre dikes“; because,
„on average, there have been only slightly more than one publication every three years (0.35 studies per year from 1925 to present). The number of studies has increased in the last three decades, with 25 publications in the last 30 years, or an average of five studies every six years.“
I do not see the causality in this statement. I think that 25 publications in 30 years on a rock type as specific as the Vredefort granophyre dikes is quite a lot. The authors can convince the reader otherwise by comparing their literature compilation with a literature compilation of e.g. the melt sheet of the Sudbury structure. If there would be 2500 publications on the Sudbury impact melt from the past 30 years, then the authors are correct that the Vredefort granophyre dikes are underrepresented in the academic literature. Without making such a comparison and without putting their literature study into perspective, it cannot be concluded that a “surprisingly small number” of published academic studies are available on the Vredefort granophyre dikes.
In general, I also advice to avoid words such as “surprisingly” from statements like this. “Surprisingly” in this particular context is misleading to the readers, and it has no scientific value.
We have removed the word "Surprisingly." We have also added context related to the Sudbury Offset Dikes. Specifically, there have been more studies on the Offset Dikes in the last five years than there have been studies on the granophyre in the last 100 years. To only have 25 studies in 30 years is not much at all when taking into consideration the fact that the granophyre is the only source of information for the Vredefort melt sheet. As indicated in the discussion with reviewer 2, there have not been any comparative studies between the dykes; amongst 9 dikes, only two have been characterized in detail. It is very little.
Moreover, we have shown that the studies that do exist have fixated on particular areas, and some of the dikes have not been investigated at all. Also, one set of samples has been investigated in 5 of the 33 publications. Even though the Vredefort granophyre is a specific rock type, it is one that is critical for understanding the Vredefort impact event and impact craters in general.
2) Table 1 - The division of studies by topic as well as by methods of investigation is somewhat random. In the current version of the manuscript, analytical methods (i.e., scientific results) such as “stable isotopes“ are presented along with research topics such as “impact component” (i.e., an interpretation of scientific data). The authors can maintain the division of topics as they like, but my advice is to more clearly distinguish between analytical methods and data on one hand (geochemistry, geophysics etc) and focal points of previous studies (impactor component) on the other hand, possibly be restructuring the table.
The organization has been modified so that all of the topics are related to the data generated, not the interpretation of the data, which is presented as subtopics. We have rearranged both the tables and the text in this regard.
3) The section on isotope geochemistry needs a careful iteration! Several sentences in the current version of this section are semantically and sometimes scientifically incorrect. Some suggestions:
Line 254: Cr isotopic content --> please change to Cr isotope compositions
Line 256: W isotopic content --> please change to W isotope compositions
Line 264: δ18O and δD isotopic data --> remove „isotopic“
(...) and the δ18O in more detail. --> I don’t understand this part of the sentence, please clarify.
„Later work reported 87Rb, 86,87Sr, 147Sm, 143,144Nd, using isotopic ratios to constrain the possible assimilated components within the granophyre [27]“. --> Please carefully check this sentence, because it contains several incorrect statements. Previous work would have reported 86Sr/88Sr and 143Nd/144Nd ratios. These are the measured values and should be reported as ratios. Note also that 88Sr is in the denominator of the Sr isotope ratio, and not 87 In contrast, 87Rb and 147Sm are not measured values but Rb/Sr and Sm/Nd ratios are reported based on Rb and Sr concentrations, as measured by isotope dilution. Also, the focus of the studies that are referred to would have been variations in radiogenic isotope abundances (86Sr, 144Nd), and variations related to stable isotope fractionation. I would generally merge this information with the section on major and trace element geochemistry (isotopes were mainly used as tracers for provenance and/or geochemical mixing processes).
Thank you for these corrections, we have implemented the suggested changes.
4) The authors present a thorough literature compilation and summarize the results of previous publications, but the overall discussion of these previous results is really thin. It is pointed out in the manuscript that the latest review paper on the Vredefort granophyre dikes did not include the bulk of the published data. I would recommend to include a more thorough scientific discussion of the literature - this is a big chance for the authors to increase the impact of the paper.
We have greatly increased the discussion of previous works substantially throughout the manuscript, including adding tables and figures, as well as adding recommendations of areas that have not seen sufficient analysis.
Round 2
Reviewer 2 Report
Dear Matthew and Elizaveta,
Thank you for your considered response to my review on "Identifying gaps in the investigation of the Vredefort granophyre dikes: A systematic literature review".
I am very pleased by the improvements to the results section of your manuscript and I now believe that the manuscript can be published without any changes.
Upon reading, I noticed a couple of items that, at your discretion, you may wish to revise before final publication. Please see below for details.
Yours,
Dr. Auriol Rae
L99-101: Isn't "shock-related ultracataclasis and melting" the same as "frictional melting coincident with the passage of the shock wave". I am unclear whether you are alluding to the ideas of Wenk (1978) whereby pseudotachylites are actually ultra-cataclasites, as opposed to a quenched melt (Maddock, 1983). The work cited (Spray, 1995) demonstrated there is no such dichotomy, that “pseudotachylyte is the result of both fracture [and] fusion” (and that comminution does not impede frictional melting as had been previously suggested). It is worth noting though, that Spray was regarding friction as the only process that generates pseudotachylites, and distinguished them from "shock veins", which if they are not frictional melts, are the consequence of decompression melting (and therefore fit neither of your descriptions). Regardless, perhaps the sentence could be clarified?
L132: Less than 20% of the publications in this review come from Earth Science, Geology, Quaternary Science, or Geosciences journals. Are there any systematic review papers closer to geological sciences? I would consider tourism sciences/human geography to be quite far removed from the area of science in this study.
L175-177: I don't believe that these two "pulses" have any statistical significance. I would remove the sentence as your next statement on the increase in publications from ~1989 onwards seems far more robust.
Reviewer 3 Report
The manuscript improved significantly compared to the previous version! I have no further comments and think that it is ready now for publication in Geosciences.